# Relationship between chromatin configuration and maturation ability of rat oocytes in vitro and in vivo

Zhaoqing Gong[1,2‡], Yujie Wang[2‡], Jiayi Tang[2‡], Yang Xu[2‡], Hongkai Wang[2‡], Yimiao Zhang[2], Lixin Xiong[2], Changzheng Sun[2], Yiyang Li[2], Yan Yang[3], Minhua Yao[2], Heng Cai[2], Zengshuo Man[2], Siyu Xuan[2], Yangyang Tang[2], Ziao Zhao[2], Jiaxin Sun[2], Dongwei Liu[2], Yanping Su[2], Xinghua Xu[2]*, Mingjiu Luo[4]*, Hongshu Sui[2]*

1 College of Basic Medical Science, Hebei University, Baoding, Heibei Province, P. R China, 2 Department of Histology and Embryology, School of Clinical and Basic Medicine, Shandong First Medical University & Shandong Academy of Medical Science, Jinan, Shandong Province, P. R China, 3 Morphological Laboratory, School of Clinical and Basic Medicine, Shandong First Medical University & Shandong Academy of Medical Science, Jinan, Shandong Province, P. R China, 4 Shandong Provincial Key Laboratory of Animal Biotechnology and Disease Control and Prevention, College of Animal Science and Veterinary Medicine, Shandong Agricultural University, Taian, Shandong Province, P. R China

‡ ZG, YW, JT, YX and HW contributed equally to this work and should be regarded as co-first authors.
* hssui@sdfmu.edu.cn (HS); luomj@sdau.edu.cn (ML); xhxu@sdfmu.edu.cn (XX)

**Data Availability Statement:** All relevant data are within the paper and its Supporting Information files.

## Abstract

### Purpose

Embryo engineering requires a large number of oocytes, which undergo in vitro maturation (IVM). Understanding how to select the best quality oocytes is key to improving IVM efficiency. Oocytes have different germinal vesicle (GV) chromatin configurations, which may explain the heterogeneity in oocyte quality during IVM. However, no reports have categorized, the chromatin configuration of rat GVs or evaluated, the association between the chromatin configuration and oocytes development.

### Methods

The GV chromatin configuration of rat oocytes was divided into seven types according to the degree of chromatin compaction: non-surrounded nucleolus (NSN), prematurely condensed NSN, partly NSN, partly surrounded nucleolus (SN-1), SN-1, condensed SN-1, and aggregated (SN-2). The chromatin configuration distribution was compared during the different stages of oocyte growth and maturation. We also analyzed the changes in the chromatin configuration at different GV stages during IVM. Moreover, the factors affecting the chromatin configuration were analyzed.

### Results

The SN-2 configuration increased with rat oocyte growth and maturation, suggesting that SN-2 facilitates oocyte development. RNA transcription activity in rat oocyte GVs was inversely correlated with oocyte IVM.

**Funding:** This work was supported by the National Natural Science Foundation of China (grant numbers 32072738 to M.L. and 81670004 to H. S.); Shandong Province Natural Science Foundation (grant number ZR2023MH267 to X.X.); and the Doctoral Startup Fund of Shandong First Medical University (grant number 001003053 to X. X.). The funders had no role in the study design, data collection and analysis, decision to publish, or preparation of the manuscript.

**Competing interests:** The authors have declared that no competing interests exist.

## Conclusions

The SN-2 chromatin configuration was related to rat oocyte growth and maturation. RNA transcription activity in rat oocytes in the GV stage was inversely correlated with oocyte maturation.

## Introduction

With the development of embryo engineering technology, important advances have been made in mammalian cloning, gene knockout, and transgenic animals, but the success rates of these techniques are low [1–5]. Therefore, the process of oocyte in vitro maturation (IVM) has received substantial research attention. IVM is an in vitro fertilization technique used to collect immature oocytes from antral follicles, with the final stages of meiosis completed during in vitro culture [6]. Although IVM is routinely used for the in vitro production of embryos in domestic species, especially cattle, its clinical use in human assisted reproduction is still evolving.

Previous studies have shown that the chromatin configuration within the germinal vesicles (GVs) of oocytes undergoes dynamic changes in some mammals during IVM, and GV break-down (GVBD) eventually occurs, after which time oocytes re-enter meiosis. These phenomena have been reported in many animals [7–9]. GV chromatin is initially dispersed, but as oocytes develop, the chromatin condenses into different structures, termed chromatin configurations. The transition between chromatin configurations marks the ability of oocytes to mature and develop.

Previous studies have analyzed the chromatin configurations of oocytes from many species, including humans [10, 11], monkeys [12], mice [13–16], rats [17], horses [18, 19], goats [20], pigs [8, 21], rabbits [22], and cows [21, 23, 24]. These studies have shown that the chromatin configuration in oocytes changes throughout the GV stages. In mammalian oocytes, GV chromatin is initially decondensed, but it condenses into different configurations with oocyte growth. For instance, with the growth of oocytes in the GV stage in mice, the chromatin configuration changes from the non-surrounded nucleolus (NSN) configuration with diffuse chromatin to the surrounded nucleolus (SN) configuration with condensed chromatin that is particularly confined around the nucleolus [15]. Other studies have shown that oocytes collected from the same ovary, and even the same follicle, can have different GV chromatin configurations in the diplotene phase of the first meiotic prophase [25]. Therefore, as well as changes in the chromatin configuration throughout the GV stages, there is substantial heterogeneity among oocytes in terms of their chromatin configurations.

It has been shown that the chromatin configuration is related to their developmental ability of mouse oocytes [16]. Moreover, in pig oocytes, changes in the chromatin configuration significantly affect follicle size, thereby affecting meiosis and oocyte developmental ability [26]. Moreover, the configuration of GV chromatin is correlated with transcriptional activity. In mice, oocytes with the SN chromatin configuration often exhibit polymerase-I/polymerase II-dependent transcriptional silencing, whereas oocytes with the NSN chromatin configuration exhibit higher transcriptional activity [27, 28]. In pigs [29], humans [30], and cows [31], it appears that transcriptional activity stops when chromatin begins to configure into a ring around the nucleolus. Another study showed that remodeling of chromatin into the SN configuration is not necessary to stop transcriptional activity [9]. In goats, although nucleolar rings of heterochromatin never form, follicular RNA synthesis declines dramatically with follicular

growth, and it stops in follicles that are ≥3 mm in diameter [20]. In rabbits, despite perinuclear ring formation as early as the primary follicle stage, oocyte transcriptional activity does not stop in large follicles if not stimulated by gonadotropin [22].

The results of these studies suggest that the chromatin configuration in GVs is related to the developmental ability of oocytes in various animals. Therefore, heterogeneity in the chromatin configuration may partly explain the poor developmental ability of in vitro embryos and may serve as a criterion for judging oocyte quality. Understanding how to select oocytes is key to improving IVM efficiency. However, no systematic studies on the configuration of chromatin in GVs have been reported in rats, which have similar reproductive physiology to humans. A better understanding of how the chromatin configuration in GVs from rats influences oocyte growth and development could lead to the development of strategies to select good quality oocytes, which could later be of relevance in the context of human assisted reproduction.

In this study, we evaluated changes in the chromatin configuration in the GVs of rat oocytes during growth and maturation. We also evaluated changes in RNA synthesis with different chromatin configurations, as well as the relationship between the chromatin configuration and healthy and atretic follicles from rats, both in vivo and in vitro. We evaluated the effect of follicle size, oocyte diameter, healthy atretic follicles, and oocyte culture on the chromatin configuration. Moreover, we evaluated the correlations of different chromatin configurations with oocyte developmental competence and RNA transcription.

## Materials and methods

### Culture medium and operating medium preparation

With the exception of the liquid M199 medium (#M0393, Sigma, US), all media were prepared at our laboratory. M2 medium prepared with triple-distilled water was used as the in vitro operating medium for rat oocytes. The live-cell fluorescent stain Hoechst 33342 (Y35467, Shanghai Yuanye Bio-Technology Co. Ltd.) was first dissolved in five-times distilled water to a concentration of 50 μg/mL (100×) before being aliquoted and stored at −20°C. The M2 operating medium was added to Hoechst 33342 to achieve a final concentration 10 μg/mL. The IVM culture medium was based on the 199 culture medium, with the addition of 10% fetal calf serum (#10099141C, Gibco, US), 1 IU/mL follicle-stimulating hormone (#10940097, Livzon Pharmaceutical Group Inc., China), and 5 IU/mL luteinizing hormone (#2016 110254634, ShuSheng Inc., China). The medium was filtered through 0.45-μm and 0.22-μm filter membranes (Y35467, Shanghai Yuanye Bio-Technology Co. Ltd.), stored at 4°C after aliquoting, and used within 2 weeks.

### Experimental animals

This study was approved by the Animal Care and Use Committee of Shandong First Medical University, and all experiments were performed following the guidelines and standards set by the National Institutes of Health (reference number W202111220327). Sprague–Dawley rats aged around 20 days were used as experimental animals, which weighed around 50 g and were not sexually matured. The animals were purchased from Jinan Pengyue Experimental Animal Breeding Co. Ltd. (Jinan, China). The rats were raised under strictly controlled light conditions, with 14 h of light (6:00–20:00) and 10 h of darkness (20:00–6:00). The rats had ad libitum access to food and water.

### Hormone treatment and oocyte acquisition

For Pregnant Mare Serum Gonadotropin (PMSG) treatment, we selected 20–28-day-old female Sprague–Dawley rats. At 15:00, PMSG was administered by intraperitoneal injection

(Tianjin Huafu High-tech Biotechnology Company, Tianjin, China) at a dose of 10 IU. Forty-eight hours later, human chorionic gonadotropin (hCG) (Tianjin Zhongbao Pharmaceutical Co., Ltd., Tianjin, China) was administered via intraperitoneal injection at a dose of 40 IU. The rats were euthanized by intraperitoneal injection of pentobarbital sodium (150–200 mg/kg) at various times after hCG injection. The abdominal cavity was opened, the ovaries were removed and washed three times with phosphate-buffered saline, and the fat was removed under a dissection microscope with the ovaries submerged in M2 medium. After trimming the excess fat, the ovaries were placed in pre-warmed M2 medium, and the follicles within the ovaries were punctured using a glass needle under a stereomicroscope, allowing the cumulus–oocyte complexes (COCs) to flow into the M2 medium.

### Measuring follicle and oocyte diameter and identifying healthy versus atretic follicles

The COCs were collected under a stereomicroscope using custom-made glass capillaries. The follicle diameter was measured by first marking 1 mm on the stereomicroscope, and then classifying the follicles into three categories (<0.5, 0.5–1, or >1 mm). The wall of healthy follicles is dense and uniform in structure, and it is slightly pink or yellowish in color. Conversely, the wall of atretic follicles is uneven in structure and gray in color, and a dark mass can sometimes be seen inside the follicle [7, 20]. We distinguished healthy from atretic follicles according to these observations.

Some COCs were then transferred to M2 medium containing 0.1% (v/v) hyaluronidase (Y35467; Shanghai Yuanye Biotechnology Co., Ltd., Shanghai, China) and the surrounding cumulus cells were removed using custom-made glass capillary tubes. Some COCs were also retained to analyze the changes in the structure of GV chromatin during IVM. After cumulus removal, the oocyte diameter was measured by laser scanning confocal microscopy (A1R MP; Nikon, Tokyo, Japan), and the oocytes were divided into three groups according to their diameter (60–69, 70–79, or 80–89 μm).

### GV chromatin configuration classification

According to the degree and distribution of chromatin compaction in the nucleus, we proposed a classification system comprising seven chromatin configurations: NSN (chromatin diffused throughout the GV with no condensed chromatin around the nucleolus), prematurely condensed NSN (cNSN; chromatin appears in a large mass in the nucleus, and there is no chromatin agglutination around the nucleolus), partly NSN (pNSN; a small amount of chromatin has condensed near the nucleolus, and the chromatin is diffused and condensed in the nucleoplasm), partly SN (pSN-1; most of the chromatin has condensed to form a ring near the nucleolus, and the chromatin is diffused and condensed in the nucleoplasm), SN-1 (chromatin has formed a complete chromatin ring around the nucleolus, and the chromatin is dispersed and condensed in the nucleoplasm), prematurely condensed SN-1 (cSN-1; chromatin has formed a complete ring around the nucleolus and appears as larger clumps in the nucleoplasm), and aggregated (SN-2; chromatin has formed a complete ring around the nucleolus, and there is no chromatin in the nucleoplasm).

### Observing the chromatin configurations of cumulus-removed oocytes

After cumulus removal from the rat oocytes, the oocytes were transferred to M2 medium containing 10 μg/mL Hoechst 33342 (Sigma) and stained for 8 min at room temperature in the dark. The oocytes were placed on slides and flattened with coverslips. The morphology of the nucleoli and nuclear membranes was observed under a phase-contrast microscope (Nikon,

Tokyo, Japan), and the configuration of GV chromatin was evaluated under a fluorescence microscope (Nikon).

## Observing the chromatin configurations of COCs during IVM

Mature culture solution (100 μL) was added to a petri dish using a micro-drop culture system, and approximately 20 COCs were placed in each drop of culture solution and cultured in an incubator at 38.5˚C containing 5% carbon dioxide with saturated humidity [7] to observe changes in the structure of GV chromatin after IVM. The COCs were cultured for different durations so that the configuration of GV chromatin during IVM could be evaluated.

## Observing the chromatin configurations of cumulus-removed oocytes during IVM

Cumulus-removed oocytes were added to M2 medium containing 3 μg/mL Hoechst 33342 (Sigma), stained at room temperature for 8 min in the dark, and M2 microdrops were added to observe the chromatin configuration. After classifying the chromatin configuration of these oocytes, cumulus-removed oocytes of different configurations were cultured in maturation medium containing 3 μg/mL Hoechst 33342 with cumulus cell-pretreated 96-well plates to simulate the in vivo growth environment of oocytes. The changes in chromatin configuration during IVM were observed at intervals of 0.5 h.

## Detection of global RNA transcription in oocytes

COCs were labeled in 100 μL 199 medium containing 1 mM 5-ethyluridine (EU) for 2 h in an incubator at 38.5˚C with 5% carbon dioxide. After EU labeling, all EU detection steps were performed at room temperature according to the manufacturer's instructions (Click-iT RNA Imaging Kit, Invitrogen, Oregon, US). After removing the cumulus and zona pellucida of the oocytes, the oocytes were fixed with 4% paraformaldehyde for 40 min. After washing the oocytes with phosphate-buffered saline, the membrane was permeabilized with 0.1% Triton X-100 for 30 min. Under dark conditions, the oocytes were treated with 100 mM Tris (pH 8.5), 1 mM copper sulfate, 10–50 μM fluorescent azide, and 100 mM ascorbic acid for 30 min. After the oocytes were washed with Click-iT reaction rinsing buffer, they were stained with Hoechst 33342 to visualize the DNA. The stained oocytes were placed on glass slides and imaged under a fluorescence microscope (Nikon). Hoechst 33342 and Alexa Fluor 488 azide were excited with a blue diode (405 nm) and an argon laser tube (488 nm), respectively.

## Statistical analysis

The independent-samples t-test was used to compare two groups. For comparisons among more than two groups, the data were first converted by least significant difference (percentage t-test), and analysis of variance was used to compare the groups, followed by Duncan's multiple-comparisons test. The results are expressed as the mean ± standard error. $P < 0.05$ was considered statistically significant.

## Results

### Configuration of GV chromatin during oocyte growth

After staining the rat oocytes with Hoechst 33342, the configuration of GV chromatin was evaluated under a fluorescence microscope. **Fig 1** illustrates the seven GV chromatin configurations in rat oocytes.

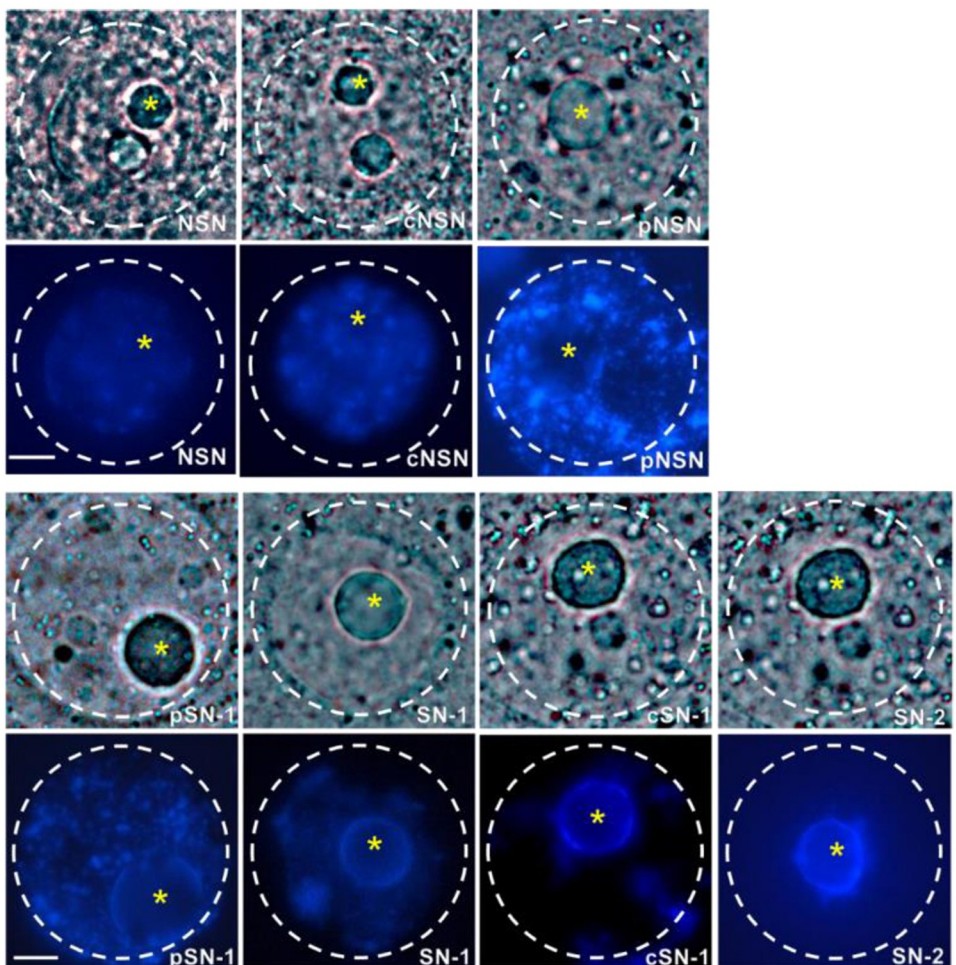

**Fig 1. Classification of the chromatin configuration in the GVs of rat oocytes.** Seven chromatin configurations were observed on light microscopy and fluorescence microscopy (NSN, cNSN, pNSN, pSN-1, SN-1, cSN-1, and SN-2). The white asterisks represent the positions of the nucleoli. Scale bar: 20 μm. GV: germinal vesicle, cNSN: prematurely condensed non-surrounded nucleolus, cSN-1: prematurely condensed surrounded nucleolus, NSN: non-surrounded nucleolus, pNSN: partly non-surrounded nucleolus, pSN-1: partly surrounded nucleolus, SN-1: surrounded nucleolus, SN-2: aggregated.

**Table 1** shows the proportions of oocytes with each chromatin configuration when the oocytes were divided into groups by follicle diameter. The proportion of oocytes with the NSN and SN-1 configurations in large follicles was significantly lower than the respective proportions in small follicles, while the proportion of oocytes with the SN-2 configuration was significantly higher in large follicles than in small follicles. This indicates that with follicular growth, the number of oocytes with chromatin in the SN-2 configuration increases, and the GV chromatin shifts toward the configuration with chromatin aggregation around the nucleolus.

**Table 2** summarizes the changes in the configuration of GV chromatin in rat oocytes with different diameters. Most oocytes with diameters of 60–69 μm had the NSN chromatin configuration. The pSN-1, SN-1, cSN-1, and SN-2 configurations appeared after the oocyte diameter was >70 μm (accounting for 76.67%). With follicular growth, the proportions of oocytes with the SN-1, cSN-1, and SN-2 configurations gradually increased, while the proportions of oocytes with the NSN, cNSN, and pNSN configurations gradually decreased. The proportion

**Table 1. Chromatin configuration in the GVs of oocytes from rat follicles of different diameters.**

| Follicle diameter (mm) | No. of oocytes | Proportion of oocytes with each chromatin configuration (%) | | | | | | | | |
|---|---|---|---|---|---|---|---|---|---|---|
| | | NSN (Total) | | | | SN (Total) | | | | |
| | | Total | NSN | cNSN | pNSN | Total | pSN-1 | SN-1 | cSN-1 | SN-2 |
| <0.5 | 357 | 15.18 ± 3.37ᵃ | 11.07 ± 3.86ᵃ | 1.02 ± 0.56ᵃ | 3.09 ± 0.86ᵃ | 84.82 ± 3.37ᵃ | 5.65 ± 1.82ᵃ | 29.89 ± 5.86ᵃ | 24.56 ± 3.81ᵃᵇ | 24.72 ± 8.80ᵃ |
| 0.5–1 | 232 | 5.44 ± 1.92ᵇ | 2.72 ± 0.89ᵇ | 0.00 ± 0.00ᵃ | 2.71 ± 1.39ᵃ | 94.56 ± 1.92ᵇ | 3.14 ± 1.68ᵃᵇ | 14.87 ± 1.13ᵇ | 32.34 ± 4.05ᵇ | 44.22 ± 4.80ᵃ |
| >1 | 90 | 0.00 ± 0.00ᵇ | 0.00 ± 0.00ᵇ | 0.00 ± 0.00ᵃ | 0.00 ± 0.00ᵃ | 100.00 ± 0.00ᵇ | 0.00 ± 0.00ᵇᵇ | 15.44 ± 4.61ᵇ | 18.56 ± 3.90ᵃ | 66.00 ± 5.52ᵇ |

COCs: cumulus–oocyte complexes, cNSN: prematurely condensed non-surrounded nucleolus, cSN-1: prematurely condensed surrounded nucleolus, GVs: germinal vesicles, NSN: non-surrounded nucleolus, pNSN: partly non-surrounded nucleolus, pSN-1: partly surrounded nucleolus, SN-1: surrounded nucleolus, SN-2: aggregated. ᵃ and ᵇ: There are significant differences between items with different letters in the same column (P < 0.05). Each treatment was replicated 3–4 times, and each replicate included approximately 30 COCs.

of oocytes with the pSN-1 configuration decreased from 15.55% at 70–79 μm to 6.47% at 80–89 μm. This indicates that GV chromatin shifts toward the configuration with chromatin aggregation around the nucleolus with oocyte growth.

## Effect of PMSG stimulation on chromatin configuration in rat oocytes

After stimulation with PMSG for 46–48 h, the oocytes were counted, stained, and observed in rats aged 20–28 days (Table 3). After PMSG treatment, the ratio of oocytes with the pSN-1 and cSN-1 configurations was lower than in the control group (without PSMG stimulation), and the ratio of oocytes with the SN-1 and SN-2 configurations was higher than in the control group. Therefore, PMSG promoted the transition of GV chromatin toward the configuration with chromatin aggregation around the nucleolus, as well as transformation of cSN-1 toward the SN-2 configuration.

## Changes in GV chromatin configuration during in vivo maturation of oocytes

Table 4 shows the changes in the GV chromatin configuration during in vivo maturation of rat oocytes. The pNSN, NSN, and cNSN configurations disappeared 48 h after injection of PMSG. The pSN-1 configuration disappeared after 2 h and the SN-1 configuration disappeared after 4 h of hCG treatment. After 1.5 h of hCG treatment, the proportion of oocytes with the SN-2 configuration sharply decreased, the proportion of oocytes with the cSN-1 configuration sharply decreased, and the proportion of oocytes with GVBD rapidly increased. With the extension of hCG treatment time, the proportion of oocytes with the cSN-1 and SN-2

**Table 2. Chromatin configuration in the GVs of rat oocytes with different diameters.**

| Diameter (μm) | Number of oocytes | Proportion of oocytes with each chromatin configuration (%) | | | | | | | | |
|---|---|---|---|---|---|---|---|---|---|---|
| | | NSN (Total) | | | | SN (Total) | | | | |
| | | Total | NSN | cNSN | pNSN | Total | pSN-1 | SN-1 | cSN-1 | SN-2 |
| 60–69 | 81 | 100.00 ± 0.00ᵃ | 76.67 ± 6.67ᵃ | 6.67 ± 6.67ᵃ | 16.66 ± 7.45ᵃ | 0.00 ± 0.00ᵃ | 0.00 ± 0.00ᵃ | 0.00 ± 0.00ᵃ | 0.00 ± 0.00ᵃ | 0.00 ± 0.00ᵃ |
| 70–79 | 146 | 27.53 ± 6.16ᵇ | 18.91 ± 5.29ᵇ | 2.99 ± 2.16ᵃ | 5.63 ± 2.32ᵃᵇ | 72.47 ± 6.16ᵇ | 15.55 ± 4.86ᵇ | 23.57 ± 3.63ᵇ | 13.65 ± 6.03ᵇ | 19.70 ± 7.87ᵇ |
| 80–89 | 125 | 5.17 ± 3.19ᶜ | 1.54 ± 1.54ᶜ | 1.43 ± 1.43ᵃ | 2.20 ± 1.44ᵇ | 94.83 ± 3.19ᶜ | 6.47 ± 2.79ᵃᵇ | 31.36 ± 6.93ᵇ | 22.57 ± 3.72ᵇ | 34.43 ± 7.13ᵇ |

All abbreviations are as listed in Table 1. ᵃ⁻ᶜ: There are significant differences between items with different letters in the same column (P < 0.05). Each treatment was replicated 3–4 times, and each replicate included approximately 30 COCs.

**Table 3. Effect of PMSG stimulation on chromatin configuration in rat oocytes.**

| PMSG | No. of oocytes | Proportion of oocytes with each chromatin configuration (%) | | | | | | | | |
|---|---|---|---|---|---|---|---|---|---|---|
| | | NSN (Total) | | | | SN (Total) | | | | |
| | | Total | NSN | cNSN | pNSN | Total | pSN-1 | SN-1 | cSN-1 | SN-2 |
| − | 163 | $10.90 \pm 2.10^a$ | $4.46 \pm 1.26^a$ | $0.00 \pm 0.00^a$ | $6.45 \pm 2.00^a$ | $89.10 \pm 2.10^a$ | $5.54 \pm 2.15^a$ | $4.66 \pm 2.40^a$ | $56.61 \pm 0.88^a$ | $22.28 \pm 1.08^a$ |
| + | 345 | $4.31 \pm 2.32^b$ | $2.17 \pm 1.09^a$ | $0.00 \pm 0.00^a$ | $2.14 \pm 1.32^a$ | $95.69 \pm 2.32^b$ | $0.59 \pm 0.33^a$ | $11.27 \pm 5.01^a$ | $23.41 \pm 2.96^b$ | $60.42 \pm 4.76^b$ |

PMSG: Pregnant Mare Serum Gonadotropin. All other abbreviations are as listed in Table 1.

[a] and [b]: There are significant differences between items with different letters in the same column (P < 0.05). Each treatment was replicated 3–4 times, and each replicate included approximately 30 COCs.

configurations gradually decreased, while the proportion of oocytes with GVBD gradually increased. These results indicate that during oocyte maturation, the cSN-1 and SN-2 configurations transformed into GVBD.

## Changes in the GV chromatin configuration during IVM of oocytes

Rat oocytes were cultured in 199 medium for different durations to evaluate changes in the GV chromatin configuration. Follicles with a diameter of 0.5 mm to >1 mm (**Table 5**) were collected during IVM. The NSN, cNSN, pNSN, and pSN-1 configurations disappeared in follicles with diameters of 0.5–1 mm, and SN-1 disappeared after 1.5 h of IVM. The proportion of oocytes with the cSN-1 configuration decreased gradually with an increase in the culture time, while the proportion of oocytes with the SN-2 configuration decreased gradually and sharply after 1 h of IVM. GVBD began to appear and increased sharply after 1 h of IVM.

In follicles with a diameter >1 mm (**Table 6**), the NSN, cNSN, pNSN, and pSN-1 configurations disappeared. The SN-1 configuration disappeared after 0.5 h of IVM. The cSN-1 configuration decreased gradually with an increase in the culture time. The SN-2 configuration increased after 0.5 h, decreased gradually, and then decreased sharply after 1 h. GVBD began

**Table 4. Changes in the configuration of GV chromatin during in vivo maturation of oocytes.**

| hCG (h) | No. of oocytes | Proportion of oocytes with each chromatin configuration (%) | | | | | | | | | |
|---|---|---|---|---|---|---|---|---|---|---|---|
| | | NSN (Total) | | | | SN (Total) | | | | | |
| | | Total | NSN | cNSN | pNSN | Total | pSN-1 | SN-1 | cSN-1 | SN-2 | GVBD |
| 0 | 345 | $4.31 \pm 2.32^a$ | $2.17 \pm 1.09^a$ | $0.00 \pm 0.00^a$ | $2.14 \pm 1.32^a$ | $95.69 \pm 2.32^{ab}$ | $0.59 \pm 0.33^a$ | $11.27 \pm 5.01^a$ | $23.41 \pm 2.96^{ab}$ | $60.42 \pm 4.76^a$ | $0.00 \pm 0.00^a$ |
| 0.5 | 246 | $0.00 \pm 0.00^b$ | $0.00 \pm 0.00^b$ | $0.00 \pm 0.00^a$ | $0.00 \pm 0.00^b$ | $100.00 \pm 0.00^a$ | $0.95 \pm 0.95^a$ | $13.72 \pm 4.02^a$ | $23.35 \pm 5.64^{ab}$ | $61.98 \pm 8.65^a$ | $0.00 \pm 0.00^a$ |
| 1 | 192 | $0.00 \pm 0.00^b$ | $0.00 \pm 0.00^b$ | $0.00 \pm 0.00^a$ | $0.00 \pm 0.00^b$ | $100.00 \pm 0.00^a$ | $0.99 \pm 0.51^a$ | $9.05 \pm 1.86^{ab}$ | $31.14 \pm 3.33^b$ | $58.82 \pm 4.86^a$ | $0.00 \pm 0.00^a$ |
| 1.5 | 233 | $0.00 \pm 0.00^b$ | $0.00 \pm 0.00^b$ | $0.00 \pm 0.00^a$ | $0.00 \pm 0.00^b$ | $99.08 \pm 0.47^a$ | $1.00 \pm 1.00^a$ | $9.24 \pm 1.48^{ab}$ | $27.46 \pm 6.16^{ab}$ | $61.38 \pm 5.79^a$ | $0.92 \pm 0.47^a$ |
| 2 | 166 | $0.00 \pm 0.00^b$ | $0.00 \pm 0.00^b$ | $0.00 \pm 0.00^a$ | $0.00 \pm 0.00^b$ | $80.96 \pm 5.75^b$ | $1.54 \pm 1.54^a$ | $6.94 \pm 1.22^{ab}$ | $32.01 \pm 7.16^b$ | $40.47 \pm 5.18^b$ | $19.04 \pm 5.75^b$ |
| 2.5 | 179 | $0.00 \pm 0.00^b$ | $0.00 \pm 0.00^b$ | $0.00 \pm 0.00^a$ | $0.00 \pm 0.00^b$ | $55.16 \pm 14.54^c$ | $0.00 \pm 0.00^a$ | $2.02 \pm 1.01^c$ | $16.47 \pm 1.93^a$ | $36.67 \pm 11.85^{bc}$ | $44.84 \pm 14.54^c$ |
| 3 | 224 | $0.00 \pm 0.00^b$ | $0.00 \pm 0.00^b$ | $0.00 \pm 0.00^a$ | $0.00 \pm 0.00^b$ | $53.78 \pm 4.35^c$ | $0.00 \pm 0.00^a$ | $3.87 \pm 0.82^{bc}$ | $29.50 \pm 2.74^{ab}$ | $20.41 \pm 5.85^{cd}$ | $46.22 \pm 4.35^c$ |
| 3.5 | 160 | $0.00 \pm 0.00^b$ | $0.00 \pm 0.00^b$ | $0.00 \pm 0.00^a$ | $0.00 \pm 0.00^b$ | $50.64 \pm 2.80^{cd}$ | $0.00 \pm 0.00^a$ | $2.26 \pm 1.16^{bc}$ | $29.16 \pm 4.37^{ab}$ | $19.22 \pm 3.30^{cd}$ | $49.36 \pm 2.80^{cd}$ |
| 4 | 126 | $0.00 \pm 0.00^b$ | $0.00 \pm 0.00^b$ | $0.00 \pm 0.00^a$ | $0.00 \pm 0.00^b$ | $41.49 \pm 2.76^{cd}$ | $0.00 \pm 0.00^a$ | $0.00 \pm 0.00^c$ | $28.68 \pm 3.09^{ab}$ | $12.81 \pm 0.85^d$ | $58.51 \pm 2.76^{cd}$ |
| 4.5 | 138 | $0.00 \pm 0.00^b$ | $0.00 \pm 0.00^b$ | $0.00 \pm 0.00^a$ | $0.00 \pm 0.00^b$ | $34.82 \pm 2.46^{de}$ | $0.00 \pm 0.00^a$ | $0.00 \pm 0.00^c$ | $24.53 \pm 1.42^{ab}$ | $10.29 \pm 2.05^d$ | $65.18 \pm 2.46^{de}$ |
| 5 | 98 | $0.00 \pm 0.00^b$ | $0.00 \pm 0.00^b$ | $0.00 \pm 0.00^a$ | $0.00 \pm 0.00^b$ | $24.25 \pm 3.42^e$ | $0.00 \pm 0.00^a$ | $0.00 \pm 0.00^c$ | $19.17 \pm 3.63^{ab}$ | $5.08 \pm 0.90^d$ | $75.75 \pm 3.42^e$ |

GVBD: germinal vesicle breakdown. All other abbreviations are as listed in Table 1.

[a–e]: There are significant differences between items with different letters in the same column (P < 0.05). Each treatment was replicated 3–4 times, and each replicate included approximately 30 COCs.

**Table 5. Changes in the GV chromatin configuration during IVM of rat oocytes from follicles with a diameter of 0.5–1 mm.**

| Culture time (h) | No. of oocytes | Proportion of oocytes with each chromatin configuration (%) | | | | | | | | | |
|---|---|---|---|---|---|---|---|---|---|---|---|
| | | NSN (Total) | | | | SN (Total) | | | | | |
| | | Total | NSN | cNSN | pNSN | Total | pSN-1 | SN-1 | cSN-1 | SN-2 | GVBD |
| 0 | 123 | 0.00 ± 0.00$^a$ | 0.00 ± 0.00$^a$ | 0.00 ± 0.00$^a$ | 0.00 ± 0.00$^a$ | 100.00 ± 0.00$^a$ | 0.00 ± 0.00$^a$ | 5.19 ± 1.65$^a$ | 49.99 ± 2.02$^a$ | 44.82 ± 0.43$^a$ | 0.00 ± 0.00$^a$ |
| 0.5 | 111 | 0.00 ± 0.00$^a$ | 0.00 ± 0.00$^a$ | 0.00 ± 0.00$^a$ | 0.00 ± 0.00$^a$ | 100.00 ± 0.00$^a$ | 0.00 ± 0.00$^a$ | 6.86 ± 1.15$^a$ | 47.72 ± 3.73$^a$ | 45.42 ± 3.22$^a$ | 0.00 ± 0.00$^a$ |
| 1 | 148 | 0.00 ± 0.00$^a$ | 0.00 ± 0.00$^a$ | 0.00 ± 0.00$^a$ | 0.00 ± 0.00$^a$ | 76.24 ± 10.64$^b$ | 0.00 ± 0.00$^a$ | 4.59 ± 0.30$^a$ | 32.33 ± 0.51$^b$ | 39.32 ± 10.60$^a$ | 23.76 ± 10.64$^b$ |
| 1.5 | 170 | 0.00 ± 0.00$^a$ | 0.00 ± 0.00$^a$ | 0.00 ± 0.00$^a$ | 0.00 ± 0.00$^a$ | 46.64 ± 7.30$^c$ | 0.00 ± 0.00$^a$ | 0.00 ± 0.00$^b$ | 27.06 ± 5.37$^b$ | 19.58 ± 1.99$^b$ | 53.36 ± 7.30$^c$ |
| 2 | 100 | 0.00 ± 0.00$^a$ | 0.00 ± 0.00$^a$ | 0.00 ± 0.00$^a$ | 0.00 ± 0.00$^a$ | 23.16 ± 6.10$^d$ | 0.00 ± 0.00$^a$ | 0.00 ± 0.00$^b$ | 16.40 ± 4.12$^c$ | 6.75 ± 2.39$^b$ | 76.84 ± 6.09$^d$ |

GVBD: germinal vesicle breakdown, IVM: in vitro maturation. All other abbreviations are as listed in Table 1.

[a–d]: There are significant differences between items with different letters in the same column (P < 0.05). Each treatment was replicated 3–4 times, and each replicate included approximately 30 COCs.

to appear and increased sharply 1 h later. These results indicate that during oocyte maturation, the cSN-1 and SN-2 chromatin configurations shift toward GVBD.

## Relationship between follicle size and oocyte meiosis

Tables 7 and 8 show the distribution of follicles of different sizes at the meiosis stage after 16 h of IVM without and with PMSG treatment. The proportion of oocytes with a follicular diameter <0.5 mm that reached the MII stage increased from 59.26% to 76.48% after PMSG treatment. The proportion of oocytes with a follicular diameter of 0.5–1 mm that reached the MII stage increased from 82.16% to 87.83% after PMSG treatment. The proportion of oocytes with a follicular diameter of >1 mm that reached the MII stage increased from 93.27% to 96.34%.

## IVM ability of rat oocytes with different chromatin configurations

S1 Table shows that among the seven chromatin configurations, oocytes with the NSN, cNSN, and pNSN configurations did not develop to the MII stage. Moreover, 28.94% of oocytes with the pSN-1 configuration, 53.62% with the SN-1 configuration, and 77.16% with the cSN-1 configuration were mature oocytes. The highest maturity rate was observed for oocytes with the SN-2 configuration (94.22%). The above results confirm that rat oocytes with the SN-2 chromatin configuration had the highest developmental ability.

**Table 6. Changes in the GV chromatin configuration during IVM of rat oocytes from follicles with a diameter of >1 mm.**

| Culture time (h) | Number of oocytes | Proportion of oocytes with each chromatin configuration (%) | | | | | | | | | |
|---|---|---|---|---|---|---|---|---|---|---|---|
| | | NSN (Total) | | | | SN (Total) | | | | | |
| | | Total | NSN | cNSN | pNSN | Total | pSN-1 | SN-1 | cSN-1 | SN-2 | GVBD |
| 0 | 199 | 0.00 ± 0.00$^a$ | 0.00 ± 0.00$^a$ | 0.00 ± 0.00$^a$ | 0.00 ± 0.00$^a$ | 100.00 ± 0.00$^a$ | 0.00 ± 0.00$^a$ | 4.12 ± 2.17$^a$ | 52.70 ± 0.70$^a$ | 43.18 ± 1.48$^a$ | 0.00 ± 0.00$^a$ |
| 0.5 | 106 | 0.00 ± 0.00$^a$ | 0.00 ± 0.00$^a$ | 0.00 ± 0.00$^a$ | 0.00 ± 0.00$^a$ | 100.00 ± 0.00$^a$ | 0.00 ± 0.00$^a$ | 0.00 ± 0.00$^b$ | 40.02 ± 2.42$^b$ | 59.98 ± 2.42$^b$ | 0.00 ± 0.00$^a$ |
| 1 | 71 | 0.00 ± 0.00$^a$ | 0.00 ± 0.00$^a$ | 0.00 ± 0.00$^a$ | 0.00 ± 0.00$^a$ | 78.79 ± 6.31$^b$ | 0.00 ± 0.00$^a$ | 0.00 ± 0.00$^b$ | 27.78 ± 5.82$^c$ | 51.01 ± 3.54$^b$ | 21.21 ± 6.31$^b$ |
| 1.5 | 103 | 0.00 ± 0.00$^a$ | 0.00 ± 0.00$^a$ | 0.00 ± 0.00$^a$ | 0.00 ± 0.00$^a$ | 30.58 ± 1.94$^c$ | 0.00 ± 0.00$^a$ | 0.00 ± 0.00$^b$ | 15.18 ± 1.73$^d$ | 15.40 ± 2.73$^d$ | 69.42 ± 1.94$^c$ |
| 2 | 80 | 0.00 ± 0.00$^a$ | 0.00 ± 0.00$^a$ | 0.00 ± 0.00$^a$ | 0.00 ± 0.00$^a$ | 19.08 ± 0.92$^d$ | 0.00 ± 0.00$^a$ | 0.00 ± 0.00$^b$ | 13.60 ± 0.87$^d$ | 5.48 ± 1.34$^e$ | 80.92 ± 0.92$^d$ |

GVBD: germinal vesicle breakdown, IVM: in vitro maturation. All other abbreviations are as listed in Table 1.

[a–e]: There are significant differences between items with different letters in the same column (P < 0.05). Each treatment was replicated 3–4 times, and each replicate included approximately 30 COCs.

**Table 7. Oocyte meiosis ability (without hormones).**

| Follicle diameter (mm) | Number of oocytes | GV | MI | AnaI/TelI | MII |
|---|---|---|---|---|---|
| <0.5 | 166 | 34.81 ± 2.67[a] | 5.18 ± 3.23[a] | 0.74 ± 0.74[a] | 59.26 ± 1.96[a] |
| 0.5–1 | 246 | 12.99 ± 0.51[b] | 4.07 ± 0.83[a] | 0.78 ± 0.78[a] | 82.16 ± 1.82[b] |
| >1 | 96 | 6.73 ± 3.64[b] | 0.00 ± 0.00[a] | 0.00 ± 0.00[a] | 93.27 ± 3.64[c] |

AnaI/TelI: anaphase I/telophase I, COCs: cumulus–oocyte complexes, GV: germinal vesicle, MI: first meiosis metaphase, MII: secondary meiosis.

[a–c]: There are significant differences between items with different letters in the same column (P < 0.05). Each treatment was replicated 3–4 times, and each replicate included approximately 30 COCs.

## Changes in the chromatin configuration during IVM of rat oocytes

Rat oocytes with the NSN configuration were cultured for different durations for IVM, and the changes in chromatin configuration were evaluated (S2 Table). Oocytes with the NSN and cNSN configurations disappeared after 2 h and 2.5 h, respectively, of IVM. The pNSN configuration began to appear after 1 h and disappeared after 5.5 h. The pSN-1 configuration began to appear after 2 h and disappeared after 13 h. The SN-1 configuration began to appear after 2 h, the cSN-1 configuration began to appear after 8.5 h, the SN-2 configuration appeared after 12.5 h, and GVBD did not appear during IVM of oocytes with the NSN configuration.

Rat oocytes with the cNSN configuration were cultured for different durations for IVM, and the changes in the chromatin configuration were evaluated (S3 Table). Oocytes with the cNSN and pNSN configurations disappeared after 2 h and 4.5 h, respectively, of IVM. The pSN-1 configuration began to appear after 1 h and disappeared after 7.5 h. The SN-1 configuration began to appear after 1 h and disappeared after 13.5 h. The cSN-1 and SN-2 configurations appeared after 3 h and 12 h, respectively, while GVBD did not appear during IVM of oocytes with the cNSN configuration.

Rat oocytes with the pNSN configuration were cultured for different durations for IVM, and the changes in the chromatin configuration were evaluated (S4 Table). Oocytes with the pNSN and pSN-1 configurations disappeared after 2 h and 6 h, respectively, of IVM. The SN-1 configuration began to appear after 1 h and disappeared after 7.5 h. The cSN-1 and SN-2 configuration appeared after 3 h and 4.5 h, respectively, while GVBD did not appear during IVM of oocytes with the pNSN configuration.

Rat oocytes with the pSN-1 configuration were cultured for different durations for IVM, and the changes in the chromatin configuration were evaluated (S5 Table). Oocytes with the pSN-1 and SN-1 configurations disappeared after 1.5 h and 4.5 h, respectively, of IVM. The cSN-1 configuration began to appear after 1 h and disappeared after 14.5 h. The SN-2 configuration appeared after 3.5 h. GVBD began to appear after 13.5 h of IVM of oocytes with the pSN-1 configuration.

**Table 8. Oocyte meiosis ability (after hormone administration).**

| Follicle diameter (mm) | Number of oocytes | GV | MI | AnaI/TelI | MII |
|---|---|---|---|---|---|
| <0.5 | 142 | 15.13 ± 0.60[a] | 4.30 ± 1.57[a] | 4.09 ± 1.57[a] | 76.48 ± 2.24[a] |
| 0.5–1 | 226 | 7.42 ± 2.00[b] | 2.89 ± 1.87[a] | 1.86 ± 1.01[ab] | 87.83 ± 0.53[b] |
| >1 | 129 | 3.05 ± 0.37[c] | 0.61 ± 0.61[a] | 0.00 ± 0.00[b] | 96.34 ± 0.92[c] |

AnaI/TelI: anaphase I/telophase I, COCs: cumulus–oocyte complexes, GV: germinal vesicle, MI: first meiosis metaphase, MII: secondary meiosis.

[a–c]: There are significant differences between items with different letters in the same column (P < 0.05). Each treatment was replicated 3–4 times, and each replicate included approximately 30 COCs.

Rat oocytes with the SN-1 configuration were cultured for different durations for IVM, and the changes in the chromatin configuration were evaluated (**S6 Table**). Oocytes with the SN-1 and cSN-1 configurations disappeared after 4 h and 8 h, respectively, of IVM. The SN-2 configuration began to appear after 2 h. GVBD began to appear after 3.5 h and gradually increased during IVM of SN-1 oocytes, and the rate of GVBD reached 93.31% after 8 h.

Rat oocytes with the cSN-1 configuration were cultured for different durations for IVM, and the changes in the chromatin configuration were evaluated (**S7 Table**). The cSN-1 configuration began to appear after 2.5 h. GVBD increased significantly after 1.5 h of IVM, and the GVBD rate reached 92.36% after 3.5 h. Simultaneously, the cSN-1 and SN-2 configurations decreased, indicating that during IVM of oocytes with the cSN-1 configuration, there was a transition from the cSN-1 and SN-2 configurations toward GVBD.

Rat oocytes with the SN-2 configuration were cultured for different durations for IVM, and the changes in the chromatin configuration were evaluated (**S8 Table**). The proportion of oocytes with GVBD reached 91.93% after 2 h. The proportion of oocytes with the SN-2 configuration decreased, indicating that there was a transition from the SN-2 configuration toward GVBD during IVM of oocytes with the SN-2 configuration.

## Configuration of GV chromatin in healthy and atretic follicles

In healthy follicles, the structure of the follicle wall is dense and uniform, and it is slightly pink or yellowish in color. In atretic follicles, the follicle wall is uneven in structure, dark in color, and there may be dark clumps inside the follicle. The proportions of atretic follicles with the pSN-1, SN-1, and SN-2 configurations were lower than the proportions of healthy follicles with these configurations. Conversely, the proportions of atretic follicles with the NSN, cNSN, pNSN, and cSN-1 configurations were significantly higher than the proportions of healthy follicles with these configurations (**S9 Table**). This suggests that the oocytes of most atretic follicles undergo premature chromatin compaction, resulting in the cNSN or cSN-1 configurations.

## Relationship between chromatin configuration and transcriptional activity in rat oocytes

**Fig 2** shows the RNA transcription activity with different GV chromatin configurations in rat oocytes. Each vertical row is the same oocyte; the upper images show Hoechst 33342 staining, and the lower images show EU staining. Transcriptional activity was observed in oocytes with six of the seven chromatin configurations (NSN, cNSN, pNSN, pSN-1, SN-1, and cSN-1). Oocytes with the SN-2 configuration did not demonstrate RNA transcriptional activity.

To study the relationship between chromatin configuration and transcriptional activity, oocytes with different chromatin configurations were labeled with 5-EU to observe RNA transcription. **S10 Table** shows that 100% of all oocytes with the NSN and cNSN configurations had RNA transcriptional activity. However, no transcriptional activity was detected in oocytes with the SN-2 configuration. Moreover, 96.88% of oocytes with the pNSN configuration, 72.16% with the pSN-1 configuration, 27.18% with the SN-1 configuration, and 4.85% with the cSN-1 configuration showed transcriptional activity. These results confirm that the RNA transcriptional activity of rat oocytes was negatively correlated with oocyte maturity.

## Discussion

In mammalian oocytes, chromatin within the GV is initially diffuse, but it condenses into different configurations with oocyte growth. To date, the chromatin configurations in rat GVs and their influence on rat meiotic competence have not been reported, despite the widespread

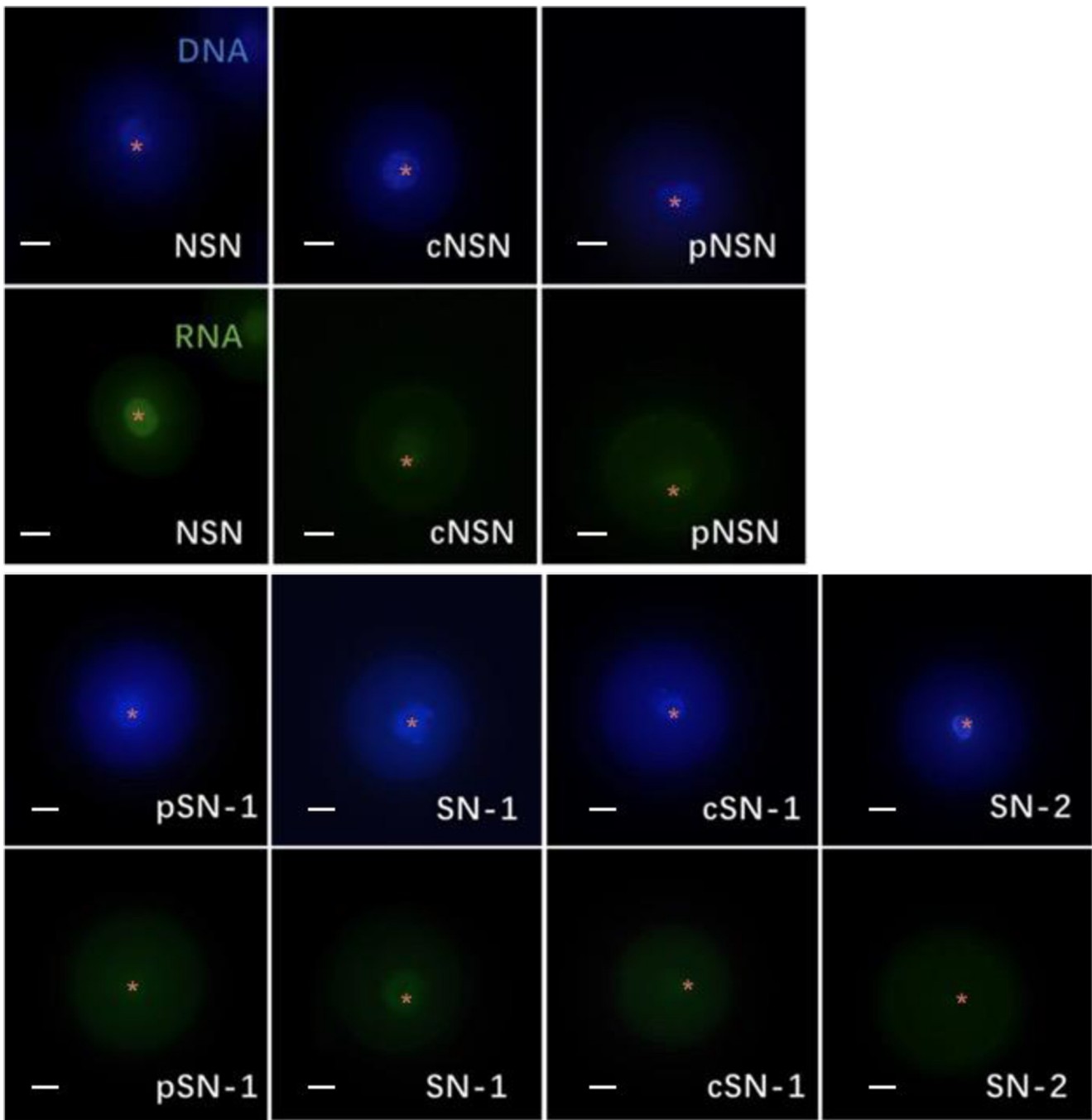

**Fig 2. Global RNA transcription of rat oocytes with different chromatin configurations.** Scale bar: 20 μm. The white asterisks represent the positions of the nucleoli. cNSN: prematurely condensed non-surrounded nucleolus, cSN-1: prematurely condensed surrounded nucleolus, NSN: non-surrounded nucleolus, pNSN: partly non-surrounded nucleolus, pSN-1: partly surrounded nucleolus, SN-1: surrounded nucleolus, SN-2: aggregated.

use of in vivo rat models to study human disease. In this study, we stained rat oocytes with Hoechst 33342 to classify the chromatin configuration of rat oocytes into seven types based on the extent of chromatin agglutination around the nucleolus and its distribution in the nucleoplasm, as follows: NSN, cNSN, pNSN, pSN-1, SN-1, cSN-1, and SN-2.

In most mammals (except goats), chromatin within GVs condenses from a diffuse state to a different configuration as the follicle grows. In this study, we evaluated the distribution of GV chromatin configurations in follicles of different sizes and in oocytes of different diameters from rats. We found that with oocyte growth and maturation, chromatin in the GV gradually condensed, transitioned from the NSN configuration to the SN-2 configuration, and formed a distinct perinuclear ring around the nucleolus.

We also measured the IVM ability of rat oocytes with different chromatin configurations. All oocytes with the NSN, cNSN, and pNSN configurations were immature. Overall, 28.94% of the oocytes with the pSN-1 configuration, 53.62% with the SN-1 configuration, and 77.16% with the cSN-1 configuration were mature. The highest maturity rate was observed for oocytes with the SN-2 configuration (94.22%), which suggests that oocytes in which chromatin fully surrounded the nucleolus had greater developmental ability than those in which chromatin only partly surrounded the nucleolus. Zuccotti found that mouse antral follicles containing oocytes with the SN configuration and the NSN configuration reached the MII stage after IVM, while oocytes with the SN configuration reached the MII stage after IVM and in vitro fertilization [16]. After oocytes with the NSN configuration matured in vitro, the embryos obtained by in vitro fertilization only developed to the two-cell stage, indicating that mouse oocytes with the SN configuration have complete developmental ability, consistent with our results in rats.

Rat oocytes demonstrated an increase in the degree of chromatin compaction in vivo and in vitro, and synchronization of the chromatin configuration was observed before ovulation. In this study, the SN-2 configuration seemed to be responsible for meiosis. Hoechst staining of live oocytes allowed us to analyze changes in the chromatin configuration during IVM. Rat oocytes synchronized to the SN-2 configuration before GVBD. This is similar to previous observations in pig oocytes, which synchronized to the GV1 chromatin configuration before GVBD, and the GV1 chromatin configuration (where chromatin is diffusely distributed throughout the GV) changed with the increase in GVBD [8]. In another study, bovine oocytes synchronized to the GVf chromatin configuration before ovulation [23]. Another study showed that the chromatin structure in goat oocytes was synchronized before GVBD [20]. In these animals, the chromatin configuration before GVBD synchronization was the same as the chromatin configuration before ovulation. Therefore, we believe that oocytes with the SN-2 configuration are capable of being excreted in rats.

We found that the chromatin configuration of oocytes changed during IVM, and the chromatin configuration progressed to GVBD at different rates. Oocytes with the pSN-1, SN-1, cSN-1, and SN-2 configurations underwent GVBD; however, oocytes with the SN-2 configuration were the quickest to undergo GVBD. Therefore, the SN-2 configuration can be considered as a sign of rat oocyte maturation. It follows that the chromatin configuration of rat oocytes appears to be related to their developmental ability, which is consistent with the results observed in other animals [7, 20, 21].

Embryo engineering involves IVM of oocytes. Despite intensive research, the survival rate of viable embryos produced in vitro is still far lower than that of embryos produced in vivo. At present, a common problem with in vitro embryo culture is the selection of the best quality oocytes. Studies have shown that oocytes obtained from the same ovary, and even the same follicle, have heterogeneous chromatin configurations, and different chromatin configurations affect oocyte meiosis and embryonic developmental ability [32]. A previous report showed that follicle size correlates with oocyte quality in both bovine and porcine models [33]. We evaluated the relationship between follicle size and meiotic stage and found that rat oocytes from larger follicles exhibited greater meiotic competence, especially oocytes from follicles

with a diameter of >1 mm. A recent study on guinea pig follicles proposed that the larger the follicular diameter, the greater the rate of oocyte meiosis to the MII stage [34].

We also compared the changes in the chromatin configuration between healthy and atretic follicles from rats. We found that the proportions of oocytes with the pSN-1, SN-1, and SN-2 configurations were lower in atretic follicles than in healthy follicles. Conversely, the proportions of oocytes with the NSN, cNSN, pNSN, and cSN-1 configurations were significantly higher in atretic follicles than in healthy follicles. This phenomenon indicates that most oocytes from atretic follicles retained the cNSN or cSN-1 configurations due to premature chromatin compaction. A recent study in pigs also showed that the proportions of oocytes from atretic follicles with the NSN and pNSN configurations were lower (and the proportions of oocytes from atretic follicles with the cNSN, cpNSN, and cpSN configurations were higher) than the proportions of healthy oocytes with these configurations [7], consistent with our results.

Genes are highly transcriptionally active during ovarian growth to accumulate large amounts of maternal RNA in preparation for early embryonic development. This study investigated the relationship between the GV chromatin configuration and transcriptional activity in rat oocytes. The results show that 100% of oocytes with the NSN and cNSN configurations had RNA transcriptional activity, but no transcriptional activity was detected in oocytes with the SN-2 configuration. Overall, 96.88% of oocytes with the pNSN configuration, 72.16% with the pSN-1 configuration, 27.18% with the SN-1 configuration, and 4.85% with the cSN-1 configuration were transcriptionally active. Previous studies have also reported the relationship between transcriptional activity and the chromatin configuration in oocytes from mice [28], pigs [7], and goats [20], supporting our observations. Another study showed that the larger the diameter of human oocytes, the weaker the transcriptional activity, which is due to the gradual change in the chromatin configuration with the increase in oocyte diameter. That is, the more mature the oocytes, the weaker the transcriptional activity [10], which is similar to our observations in rat oocytes.

Although three chromatin configuratins have been proposed in human oocytes, we classified the chromatin configurations in rat oocytes in more detail in this study. We verified seven types of chromtin configuration in rat oocytes by staining live cells. This study on rat oocytes provides criteria for oocyte selection based on the chromatin configuration, which may later be valuable in the context of human embryo engineering and assisted reproductive technology.

## Limitations

This study has some limitations that should be considered. First, the quality and quantity of oocytes from different rat strains may have differed; Sprague–Dawley rats tend to be outbred, leading to more interindividual bias. Moreover, the oocytes were derived from only a small number of rats, so the generalizability of the findings is limited. Second, the definitions and classifications of chromatin configurations differ between studies, making it difficult to directly compare our results with those of others. Moreover, the method of oocyte IVM and the application of hormones in the maturation medium may differ between the present study and previous studies, again making direct comparisons difficult. Future studies should investigate the effects of supplementation with different hormones on the IVM of rat oocytes.

## Conclusions

In conclusion, the present study revealed the relationship between the GV chromatin configuration and the developmental ability of rat oocytes in vitro. The highest maturity rate was

observed for oocytes with the SN-2 configuration, and this configuration seemed to be responsible for meiosis. Rat oocytes from larger follicles exhibited greater meiotic competence, especially oocytes from follicles with a diameter >1 mm. Moreover, the proportions of oocytes with the pSN-1, SN-1, and SN-2 configurations were lower in atretic follicles than in healthy follicles. These observations provide a theoretical basis for developing more appropriate oocyte selection criteria for embryo engineering and assisted reproduction, which could in turn improve the quantity and quality of IVM oocytes.

## Supporting information

**S1 Table. Maturation ability of rat oocytes with different chromatin configurations.** cNSN: prematurely condensed non-surrounded nucleolus, COC: cumulus–oocyte complexes, cSN-1: prematurely condensed surrounded nucleolus, NSN: non-surrounded nucleolus, pNSN: partly non-surrounded nucleolus, pSN-1: partly surrounded nucleolus, SN-1: surrounded nucleolus, SN-2: aggregated. [a–e]: There are significant differences between items with different letters in the same column (P < 0.05). Each treatment was replicated 3–4 times, and each replicate included approximately 20 COCs.
(DOCX)

**S2 Table. Changes in the chromatin configuration during IVM of rat oocytes with the NSN configuration.** GVBD: germinal vesicle breakdown, IVM: in vitro maturation. All other abbreviations are as listed in Table 1. [a–q]: There are significant differences between items with different letters in the same column (P < 0.05). Each treatment was replicated 3–4 times, and each replicate included approximately 15 COCs.
(DOCX)

**S3 Table. Changes in the chromatin configuration during IVM of rat oocytes with the cNSN configuration.** GVBD: germinal vesicle breakdown, IVM: in vitro maturation. All other abbreviations are as listed in Table 1. [a–q]: There are significant differences between items with different letters in the same column (P < 0.05). Each treatment was replicated 3–4 times, and each replicate included approximately 15 COCs.
(DOCX)

**S4 Table. Changes in the chromatin configuration during IVM of rat oocytes with the pNSN configuration.** GVBD: germinal vesicle breakdown, IVM: in vitro maturation. All other abbreviations are as listed in Table 1. [a–t]: There are significant differences between items with different letters in the same column (P < 0.05). Each treatment was replicated 3–4 times, and each replicate included approximately 15 COCs.
(DOCX)

**S5 Table. Changes in the chromatin configuration during IVM of rat oocytes with the pSN-1 configuration.** GVBD: germinal vesicle breakdown, IVM: in vitro maturation. All other abbreviations are as listed in Table 1. [a–r]: There are significant differences between items with different letters in the same column (P < 0.05). Each treatment was replicated 3–4 times, and each replicate included approximately 15 COCs.
(DOCX)

**S6 Table. Changes in the chromatin configuration during IVM of rat oocytes with the SN-1 configuration.** GVBD: germinal vesicle breakdown, IVM: in vitro maturation. All other abbreviations are as listed in Table 1. [a–i]: There are significant differences between items with different letters in the same column (P < 0.05). Each treatment was replicated 3–4 times, and

each replicate included approximately 30 COCs.
(DOCX)

**S7 Table. Changes in the chromatin configuration during IVM of rat oocytes with the cSN-1 configuration.** GVBD: germinal vesicle breakdown, IVM: in vitro maturation. Other abbreviations are as listed in Table 1. [a–g]: There are significant differences between items with different letters in the same column (P < 0.05). Each treatment was replicated 3–4 times, and each replicate included approximately 30 COCs.
(DOCX)

**S8 Table. Changes in the chromatin configuration during IVM of rat oocytes with the SN-2 configuration.** GVBD: germinal vesicle breakdown, IVM: in vitro maturation. Other abbreviations are as listed in Table 1. [a–d]: There are significant differences between items with different letters in the same column (P < 0.05). Each treatment was replicated 3–4 times, and each replicate included approximately 30 COCs.
(DOCX)

**S9 Table. Configuration of GV chromatin in healthy and atretic follicles from rats.** GV: germinal vesicle. All other abbreviations are as listed in Table 1. [a–b]: There are significant differences between items with different letters in the same column (P < 0.05). Each treatment was replicated 3–4 times, and each replicate included approximately 30 COCs.
(DOCX)

**S10 Table. Chromatin configuration and transcriptional activity of rat oocytes.** All abbreviations are as listed in Table 1.
(DOCX)

## Author Contributions

**Conceptualization:** Zhaoqing Gong, Yang Xu, Minhua Yao, Hongshu Sui.

**Data curation:** Zhaoqing Gong, Changzheng Sun, Siyu Xuan.

**Formal analysis:** Zhaoqing Gong, Yujie Wang, Jiayi Tang, Yang Xu, Lixin Xiong, Changzheng Sun, Yiyang Li, Yan Yang, Siyu Xuan, Ziao Zhao, Jiaxin Sun.

**Funding acquisition:** Xinghua Xu, Mingjiu Luo, Hongshu Sui.

**Investigation:** Zhaoqing Gong, Yujie Wang, Hongkai Wang, Yimiao Zhang, Lixin Xiong.

**Methodology:** Zhaoqing Gong, Jiayi Tang, Yang Xu, Lixin Xiong, Yiyang Li, Minhua Yao, Ziao Zhao, Dongwei Liu, Hongshu Sui.

**Project administration:** Xinghua Xu.

**Resources:** Zhaoqing Gong, Yujie Wang, Yang Xu, Hongkai Wang, Yimiao Zhang, Heng Cai, Zengshuo Man, Yangyang Tang, Jiaxin Sun, Dongwei Liu, Mingjiu Luo.

**Software:** Zhaoqing Gong, Hongkai Wang, Yimiao Zhang, Yan Yang, Heng Cai, Dongwei Liu, Yanping Su, Mingjiu Luo.

**Supervision:** Hongkai Wang, Heng Cai, Yanping Su.

**Validation:** Zengshuo Man, Yangyang Tang.

**Visualization:** Zengshuo Man, Yangyang Tang, Xinghua Xu.

**Writing – original draft:** Zhaoqing Gong, Yujie Wang, Jiayi Tang.

**Writing – review & editing:** Zhaoqing Gong, Changzheng Sun, Yiyang Li, Yan Yang.

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
