## [Decision Letter · Decision Letter 0]

4 Jul 2024

PONE-D-24-16827Relationship between chromatin configuration and maturation ability in vitro and in vivo in rat oocytesPLOS ONE

Dear Dr. Sui,

Thank you for submitting your manuscript to PLOS ONE. After careful consideration, we feel that it has merit but does not fully meet PLOS ONE’s publication criteria as it currently stands. Therefore, we invite you to submit a revised version of the manuscript that addresses the points raised during the review process.

We look forward to receiving your revised manuscript.

Kind regards,

Academic Editor

PLOS ONE

Journal Requirements:

https://journals.plos.org/plosone/s/file?id=ba62/PLOSOne_formatting_sample_title_authors_affiliations.pdf"

Reviewers' comments:

Reviewer's Responses to Questions

**Comments to the Author**

1. Is the manuscript technically sound, and do the data support the conclusions?

Reviewer #1: Partly

Reviewer #2: Partly

Reviewer #3: Yes

2. Has the statistical analysis been performed appropriately and rigorously? 

Reviewer #1: Yes

Reviewer #2: Yes

Reviewer #3: Yes

3. Have the authors made all data underlying the findings in their manuscript fully available?

Reviewer #1: Yes

Reviewer #2: Yes

Reviewer #3: Yes

4. Is the manuscript presented in an intelligible fashion and written in standard English?

Reviewer #1: No

Reviewer #2: Yes

Reviewer #3: Yes

5. Review Comments to the Author

Reviewer #1: The work seems to be interesting and has clinical significance.

The abstract needs to be rewritten using simple sentences.

Some other microscopic techniques can be carried out to validate the observations

Reviewer #2: The manuscript by Gong et al, titled " Relationship between chromatin configuration and maturation ability in vitro and in vivo in rat oocytes" reports changes in the chromatin configuration in the germinal vesicles (GVs) of rat oocytes regulates their growth and maturation in in vitro culture. Specifically, they show that the SN-2 chromatin configuration in GVs of the rat oocytes increased with rat oocyte growth and maturation. The authors infer from these observations that the SN2 configuration enables or “permits” oocyte development.

While the correlation is intriguing and the manuscript is well written, their conclusion is based upon conjecture. It is equally plausible that development cues within the oocyte confers the SN2 configuration within its GV chromatin.

To draw unambiguous conclusion that “SN2 configuration enables or “permits” oocyte development”, the authors must experimentally show that altering the chromatin configuration of the GVs also changes maturation rate of the Oocytes.

Reviewer #3: The manuscript entitled “Relationship between Chromatin Configuration and Maturation Ability In Vitro and In Vivo in Rat Oocytes” by Zhaoqing Gong, Yujie Wang, Jiayi Tang, Yang Xu, Hongkai Wang et al., presents a significant study in reproductive biology and in vitro embryo generation. The authors have put tremendous effort into this study, which is well appreciated. The relevance of chromatin compaction in embryo development is a well-studied topic, and the negative correlation between chromatin compaction and transcription is unarguable. Although this research is reproduced in a new model system, the study's novelty can be questioned, though no credits are taken away from this essential scientific research.

Suggestions:

1. I strongly recommend that the authors avoid number tables and substitute them with bar graphs to attract more reader attention and highlight the significance of data.

2. The authors explain the role of PMS Gonadotropin in chromatin configuration in the results, but they do not mention the oocyte size range.

3. In the results where the authors are looking for chromatin configuration during IVM, all the hours mentioned are offset by half an hour. Please clarify these observations (Table S4-S8).

4. When comparing the chromatin configuration of healthy and atretic follicles, only prematurely condensed non-surrounded nucleolus (cSN-1) seems to be higher in atretic oocytes. Even though the cSN-1 configuration is significantly higher in atretic oocytes, the SN-2 configuration seems less, indicating it is the limiting stage for development. Is there any explanation for these observations?

5. In the transcription assay, the authors should explain the zero-transcription activity in the oocyte with the SN-2 configuration, specifically how protein homeostasis is maintained without transcription.

6. In the discussion, the authors mentioned the similarity between the widespread use of rat models in human diseases. I encourage the authors to explain the similarities and differences between rat oocyte models and human systems for oocyte development. This would justify the use of rats in this study.

Overall, this is a well-documented and designed study. As mentioned, the novelty is still questioned, but explaining the system’s similarities with humans will improve the relevance of these observations.

6. PLOS authors have the option to publish the peer review history of their article (what does this mean?). If published, this will include your full peer review and any attached files.

Reviewer #1: No

Reviewer #2: No

Reviewer #3: No

---

## [Author Response · Author response to Decision Letter 0]

5 Aug 2024

Dear Editor,

We would like to take this opportunity to thank both yourself and the reviewers for considering our manuscript for publication in PLoS One. We sincerely appreciate that the scientific merit of our research is recognized, and we have exercised our best efforts to address all of the comments and suggestions received from the journal.

We hope our manuscript is now considered suitable for publication in PLoS One.

We look forward to hearing from you at your earliest convenience.

Yours sincerely,

Dr Sui

List of Actions

LOA1: Is the manuscript presented in an intelligible fashion and written in standard English? Reviewer #1: No

LOA2: The abstract needs to be rewritten using simple sentences.

Some other microscopic techniques can be carried out to validate the observations

Response: We have rewrittened the abstract using simple sentences.

LOA3: To draw unambiguous conclusion that “SN2 configuration enables or “permits” oocyte development”, the authors must experimentally show that altering the chromatin configuration of the GVs also changes maturation rate of the Oocytes.

Response: We thank the reviewer for this valuable suggestion. The S1 Table shows the maturity of the oocytes with different chromatin configurations. The data illustrate that when the chromatin configuration changes, the maturity of the oocytes also changes.

LOA4: Although this research is reproduced in a new model system, the study's novelty can be questioned

Response: We thank the reviewer for this important comment. It is true that previous studies have evaluated the germinal vesicle chromatin configuration in oocytes from other mammals, but most of these studies have lacked thoroughness. In our study, we used rats as the experimental animals, which is novel. Moreover, we divided the oocyte chromatin configurations into more detailed configurations, providing criteria for oocyte selection, which would be useful in the context of embryo engineering and assisted reproductive technology. We also stained live oocytes to verify the seven different types of chromatin configuration in foamed oocytes based on our classification, which has not been performed in other mammals. We hope this explanation clarifies the novel aspects of our study.

Responses to Suggestions

1. I strongly recommend that the authors avoid number tables and substitute them with bar graphs to attract more reader attention and highlight the significance of data.

Response: We thank the reviewer for this valuable suggestion. We agree that there is a substantial number of tables. However, if the tables are changed to graphs, many separate graphs would be needed to convey the data from each table, which will make the data messy. For example, if we convert a table in which the horizontal coordinate is the chromatin configuration and the ordinate is time, the table must be divided into at least seven graphs, as each graph can only represent one configuration. When presented this way, the overall change in the chromatin configuration cannot be observed. Moreover, it is not possible to observe the relationship between the various chromatin configurations using graph format.

2. The authors explain the role of PMS Gonadotropin in chromatin configuration in the results, but they do not mention the oocyte size range.

Response: We thank the reviewer for this suggestion. To clarify, when studying the effects of gonadotropin on the chromatin configuration, it is not necessary to distinguish the size of the oocytes/follicles. Instead, 48 hours after gonadotropin injection, oocytes were collected from the rats to evaluate the changes in chromatin configuration in response to gonadotropin injection, as shown in Table 3.

3. In the results where the authors are looking for chromatin configuration during IVM, all the hours mentioned are offset by half an hour. Please clarify these observations (Table S4-S8).

Response: We thank the reviewer for pointing this out. To observe the changes in the chromatin configuration during oocyte growth and maturation in detail, we evaluated the specific changes in the chromatin configuration at intervals of 30 minutes. We hope this clarifies why the hours mentioned are offset by half an hour.

4. When comparing the chromatin configuration of healthy and atretic follicles, only prematurely condensed non-surrounded nucleolus (cSN-1) seems to be higher in atretic oocytes. Even though the cSN-1 configuration is significantly higher in atretic oocytes, the SN-2 configuration seems less, indicating it is the limiting stage for development. Is there any explanation for these observations?

Response: We thank the reviewer for this pertinent question. The cSN-1 configuration does not represent a restricted stage of development; rather, it represents the near-mature configuration of oocytes. As the reviewer pointed out, in oocytes from atretic follicles, the cSN-1 configuration was significantly higher, while the SN-2 configuration was lower. This indicates that the oocytes should have developed into the SN-2 configuration, but due to follicular atresia, the oocytes remained more stagnant in the cSN-1 configuration.

5. In the transcription assay, the authors should explain the zero-transcription activity in the oocyte with the SN-2 configuration, specifically how protein homeostasis is maintained without transcription.

Response: We thank the reviewer for this valuable suggestion. The RNA transcription kit used in this study tested the ability of oocytes to synthesize RNA. In the SN-2 configuration, the oocytes have matured; that is, they no longer synthesize new RNA, so there is no transcriptional activity. The oocytes still have RNA; therefore, protein homeostasis is not affected.

6. In the discussion, the authors mentioned the similarity between the widespread use of rat models in human diseases. I encourage the authors to explain the similarities and differences between rat oocyte models and human systems for oocyte development. This would justify the use of rats in this study.

Overall, this is a well-documented and designed study. As mentioned, the novelty is still questioned, but explaining the system’s similarities with humans will improve the relevance of these observations.

Response: We thank the reviewer for their positive comments on our study, and we agree that elaborating on the similarities and differences between rats and humans is valuable. The chromatin configuration in human oocytes can be divided into three classes: Class A, in which chromatin is distributed throughout the germinal vesicle; Class B, in which chromatin agglutinates within the germinal vesicle; and Class C, in which chromatin further agglutinates and completely or partially surrounds large nucleoloid bodies. Class A in human oocytes resembles the NSN, cNSN, pNSN, and pSN-1 chromatin configurations in rat oocytes, Class B is similar to the SN-1 and cSN-1 configurations in rat oocytes, and Class C is similar to the SN-2 configuration in rat oocytes. Moreover, the larger the diameter of human oocytes, the weaker the transcriptional activity, which is due to the gradual change in the chromatin configuration with the increase in oocyte diameter. That is, the more mature the oocytes, the weaker the transcriptional activity, which is similar to our results in rat oocytes.

Although there are three chromatin configurations in human oocytes, we classified the chromatin configurations in more detail in rats in this study. We verified the seven types of chromatin configuration in foaming oocytes according to our classification by staining live cells. This study on rat oocytes provides criteria for oocyte selection, which is valuable in the context of human embryo engineering and assisted reproductive technology.

---

## [Decision Letter · Decision Letter 1]

4 Oct 2024

Relationship between chromatin configuration and maturation ability of rat oocytes in vitro and in vivo

PONE-D-24-16827R1

Dear Dr. Sui,

We’re pleased to inform you that your manuscript has been judged scientifically suitable for publication and will be formally accepted for publication once it meets all outstanding technical requirements.

Kind regards,

Academic Editor

PLOS ONE

Additional Editor Comments (optional):

Reviewers' comments:

Reviewer's Responses to Questions

**Comments to the Author**

1. If the authors have adequately addressed your comments raised in a previous round of review and you feel that this manuscript is now acceptable for publication, you may indicate that here to bypass the “Comments to the Author” section, enter your conflict of interest statement in the “Confidential to Editor” section, and submit your "Accept" recommendation.

Reviewer #3: All comments have been addressed

2. Is the manuscript technically sound, and do the data support the conclusions?

Reviewer #3: Yes

3. Has the statistical analysis been performed appropriately and rigorously? 

Reviewer #3: Yes

4. Have the authors made all data underlying the findings in their manuscript fully available?

Reviewer #3: Yes

5. Is the manuscript presented in an intelligible fashion and written in standard English?

Reviewer #3: Yes

6. Review Comments to the Author

Reviewer #3: (No Response)

7. PLOS authors have the option to publish the peer review history of their article (what does this mean?). If published, this will include your full peer review and any attached files.

Reviewer #3: **Yes: **vinesh vinayachandran

---

## [Editor Report · Acceptance letter]

21 Nov 2024

PONE-D-24-16827R1 

PLOS ONE

Dear Dr. Sui, 

I'm pleased to inform you that your manuscript has been deemed suitable for publication in PLOS ONE. Congratulations! Your manuscript is now being handed over to our production team.

Kind regards, 

on behalf of

Dr. Rajakumar Anbazhagan 

Academic Editor

PLOS ONE